Subject Area:
cellular biology

Keywords:
angiogenesis, bone marrow niche, osteogenesis, repair, regeneration

Author for correspondence:
Anjali P. Kusumbe
e-mail: anjali.kusumbe@kennedy.ox.ac.uk

†These authors contributed equally to this study.

# Role of angiocrine signals in bone development, homeostasis and disease

Unnikrishnan Sivan†, Jessica De Angelis† and Anjali P. Kusumbe

The Kennedy Institute of Rheumatology, University of Oxford, Oxford OX3 7FY, UK

APK, 0000-0002-1911-5495

Skeletal vasculature plays a central role in the maintenance of microenvironments for osteogenesis and haematopoiesis. In addition to supplying oxygen and nutrients, vasculature provides a number of inductive factors termed as angiocrine signals. Blood vessels drive recruitment of osteoblast precursors and bone formation during development. Angiogenesis is indispensable for bone repair and regeneration. Dysregulation of the angiocrine crosstalk is a hallmark of ageing and pathobiological conditions in the skeletal system. The skeletal vascular bed is complex, heterogeneous and characterized by distinct capillary subtypes (type H and type L), which exhibit differential expression of angiocrine factors. Furthermore, distinct blood vessel subtypes with differential angiocrine profiles differentially regulate osteogenesis and haematopoiesis, and drive disease states in the skeletal system. This review provides an overview of the role of angiocrine signals in bone during homeostasis and disease.

## 1. Introduction

The vascular system serves as a rapid transport network for delivering oxygen and nutrients. In addition to this traditional role, recent evidence illustrates that endothelial cells (ECs) and perivascular cells engage in signalling with neighbouring cells, and regulate various tissue and organ developments and functions [1–5]. These interactions between the vasculature and tissue cells involve paracrine or juxtacrine signalling, also termed as 'angiocrine signalling'. The angiocrine signals involve growth factors, extracellular matrix components, secreted signalling molecules such as cytokines and chemokines, and gaseous, physical or cell–cell communication through the cell surface molecules. During such angiocrine crosstalk with neighbouring cell types, blood vessels often form nurturing niche microenvironments required for the maintenance of stem and progenitor cells [6]. In bone, vasculature provides specialized niches for haematopoietic stem cells (HSCs) and osteoprogenitors and regulates haematopoiesis and osteogenesis [6]. This review aims to provide a summary of angiocrine factors and the role of angiogenesis in the skeletal system. It also provides an evaluation of the impact of angiocrine crosstalk on bone physiology and pathophysiology. The angiocrine factors in bone are summarized in table 1.

## 2. Niche functions of blood vessels during bone formation

The circulatory network in the mammalian skeletal system controls the development of bone through angiocrine signalling. Bone formation starts with the migration and localization of cells to a specific micro-niche followed by condensation of mesenchymal cells [34,35]. This mesenchymal condensate then acts as a template for further differentiation and development [36]. Even though vascular invasion and blood vessel growth is a later event in bone development [37], some blood vessel-derived factors/angiocrine signals from the periphery may play a

**Table 1.** Angiocrine factors and their crosstalk with tissue cells in bone.

| angiocrine factor | source | target cell | function | reference |
|---|---|---|---|---|
| OPG | endothelial cell | osteoclasts | inhibit osteoclastogenesis | [7] |
| SEMA-III | endothelial cells | osteoclasts | bone remodelling | [8–11] |
| IL-33 | CD105+ endothelial cells | osteoblasts | osteogenesis, haematopoiesis | [12] |
| BMP-2 | endothelial cells | chondrocytes | endochondral bone formation, fracture repair | [13,14] |
| matrix metalloproteinases: Mmp2, Mmp9, Mmp14 | type H endothelial cells | chondrocytes | cartilage resorption, directional bone elongation | [15] |
| Timp1, Timp2, Timp3, Timp4 | type H endothelial cells | chondrocytes | bone resorption and remodelling | [15] |
| SCF | type H, arterial and sinusoidal endothelial cells | HSCs | HSC maintenance | [16] |
| nidogen-1 | sinusoidal and perivascular stromal cells | pro-B cells | pro-B cell maintenance | [17] |
| IL-7 | endothelial cells and perivascular stromal cells | pro-B cells | pro-B cell maintenance | [18,19] |
| CXCL12 | endothelial cells and mesenchymal stem cells | HSCs | HSC maintenance | [20,21] |
| tenascin-C | endothelial cells | HSCs | HSC survival | [17] |
| FGF-2 | endothelial cells | HSPCs | HSPC expansion | [22,23] |
| Jag-1 | endothelial cells | HSCs | HSC regeneration, lymphoma cell proliferation | [24] |
| NOS2 | endothelial cells | osteoblast | negative regulation of osteoblast differentiation | [25] |
| PDGF | endothelial cells | osteoprogenitor | osteoprogenitor proliferation and survival | [26] |
| TGF | endothelial cells | osteoprogenitor | osteoprogenitor survival | [26] |
| FGF1 | endothelial cells | osteoblast and osteoprogenitor | osteoprogenitor survival | [26] |
| Noggin | endothelial cells | osteoblast and osteoprogenitor | bone growth, mineralization and chondrocyte maturation | [27] |
| BMP-4 | endothelial cells | HSPC | expansion of HSPC | [23] |
| angiopotein-1 | endothelial cells | HSPC | protection of HSPC | [22] |
| VCAM-1 | endothelial cells | osteoclasts, leucocytes and fibroblasts | leucocytes trafficking, protection of DTCs | [28–31] |
| E-selectin | endothelial cells | osteoclasts, leucocytes | trafficking leucocytes, cancer metastasis | [28–30,32] |
| von Willebrand factor | endothelial cells | disseminated tumour cells | protection of DTCs | [31] |
| thrombospondin-1 | endothelial cells | disseminated tumour cells | quiescence of DTCs | [33] |
| IGFBP2 | endothelial cells | HSPC | expansion of HSPCs | [23] |
| ICAM-1 | endothelial cells | leucocytes and fibroblasts | leucocytes trafficking | [28–30] |

role as early as during mesenchymal condensation. Transforming growth factor beta 1 (TGFβ1) upregulates the production of connective tissue growth factor (CTGF), and CTGF is a downstream effector of TGFβ1 for surrounding cells, including ECs. CTGF and TGFβ1 are found to be upregulated in mesenchymal condensations [38]. Precursors of ECs termed as 'angioblasts' are present early in organ bud formation before vascular development [2]. During limb formation, there is a prominent

expression of vascular endothelial growth factor (VEGF) in the mesenchymal condensate [39]. VEGF enhances osteogenesis and helps in the vascular patterning during osteogenesis [40]. After mesenchymal condensation, bone formation occurs by either of the two processes: (i) intramembranous ossification or (ii) endochondral ossification. During intramembranous ossification, mesenchymal cells in the condensate differentiate into osteoblasts, which then differentiate to osteocytes/bone cells to generate flat bones such as the skull and facial bones [41]. Alternatively, in endochondral ossification, long bones develop through an intermediate stage of chondrocyte differentiation and avascular cartilage formation [42].

The central player in blood vessel invasion to the bone tissue after mesenchymal condensation is hypoxia, where hypoxia-inducible factors (HIFs) signal the oxygen level [43]. Under normoxic conditions, the HIF1 subunits are targeted for proteasomal degradation by hydroxylation. While in hypoxia, due to the limiting oxygen levels for hydroxylation, the HIF1-$\alpha$ subunits are stabilized and activate downstream signalling pathways, including VEGF signalling [6,44]. In line with this, HIF1-$\alpha$ loss of function mice shows a decline in bone volume and bone vascularity [37]. VEGF signalling from the avascular regions, which have high levels of VEGF receptors, recruits ECs and drives blood vessel growth. VEGF signalling plays a central role in coupling angiogenesis and osteogenesis [45], through its effect on ECs and also by influencing chondrocytes, osteoblasts and osteoclasts [46].

During the postnatal stages, the vasculature of the skeletal system is known to play essential roles in bone growth and bone formation; however, bone vasculature remained vaguely defined as a network of arteries and sinusoidal blood vessels until recent years. Owing to its complex and calcified nature, imaging of the bone tissue remained difficult. Recent improvements with bone imaging techniques provided new insights into the organization of blood vessels and highlighted the heterogeneity of blood vessels in the skeletal system [47]. Notably, in addition to arteries, veins and sinusoidal vessels, a structurally, phenotypically and functionally distinct capillary subtype is present in bone. These capillaries localize in the metaphysis and cortical regions of bones, physically associate with osteoprogenitors and generate an active niche microenvironment for cells of the osteoblast lineage. Due to the high expression of specific markers, they are termed as type H [26]. The abundance of these type H vessels gradually declines in adult and ageing mice, which provides a compelling explanation for the age-associated loss of bone mass seen in rodents and humans. Genetic and pharmacological approaches revealed that the reactivation of type H endothelium in aged mice resulted in increased osteoprogenitor numbers and improved bone mass [48]. Particularly, endothelial Hif1-$\alpha$ maintains these vessels which diminish upon ageing. Treatment with Hif1 stabiliser in aged mice leads to the expansion of type H ECs, accumulation of surrounding osteoprogenitors and increase in bone mass and bone quality [48]. Furthermore, molecular and mechanistic analysis of angiogenesis and type H ECs indicated that these ECs mediate developmental and regenerative angiogenesis in the bone (figure 1).

ECs are known to produce Wnt5a [49]. Wnt5a is a secreted glycoprotein that mediates the beta-catenin signalling pathway, which is a central regulator of osteogenesis [50]. VEGF overexpression conditions lead to the stabilization of beta-catenin and excessive bone ossification, indicating the crosstalk between angiogenesis and bone formation via Wnt signalling [50]. Osteoblast-derived Wnt5a is a key player in growth plate ossification and an essential mediator of osteoblastic differentiation through bone morphogenetic protein 2 (BMP-2) [13]. However, the involvement of endothelium-derived Wnts in skeletal system development needs further examination. Another class of extracellular signalling molecules having a strong implication in bone formation and remodelling are semaphorins (Sema) [8,9]. For example, Sema-III is an active member of Sema family with a known role in bone patterning [10,11]. In addition to these factors, ECs secrete proteases like matrix metalloproteinases (Mmps), including Mmp2, Mmp9 and Mmp14 [15]. These Mmps upregulate type H ECs mediate cartilage resorption and bone formation with the help of vessel-associated osteoclasts (VAO), a newly discovered counterpart of bone-associated osteoclasts (BAO). The loss of Mmps in type H ECs leads to misdirected bone growth and abnormal bone elongation [15]. Thus, endothelium-derived factors play a central role in driving osteogenesis and bone growth. The role of angiocrine factors in osteogenesis is summarized and illustrated in figure 1.

# 3. Therapeutic potential of angiocrine factors in bone repair and regeneration

Unlike most other organs in the body, bone possesses a high regenerative potential. Usually, bone repair and regeneration following fracture does not form a fibrotic scar, a common phenomenon occurring during repair of soft tissues. Bone repair occurs in four stages; first, the site of fracture is encapsulated by a haematoma, establishing a hypoxic environment with significant upregulation of HIF-1$\alpha$ and VEGF [51,52]. In response to chronic hypoxia, ECs upregulate the osteogenic factor BMP-2 [14]. Noggin, a secreted BMP agonist, regulated via endothelial Notch signalling reverses both vascular and bone defects [27]. Since Notch signalling is known to play a role in fracture repair [53], there could be a possible angiocrine function via Notch signalling in fracture healing. Second, the fracture site is invaded by new angiogenic blood vessels, laying down a template for osteoclast–fibrocartilaginous callus formation, as blood vessels recruit and guide osteoblast precursors to the site of fracture [54,55]. Third, the soft callus calcifies to generate new bone, which requires early and prolonged exposure to exogenous VEGF to promote vascularization and bone growth. Blocking endogenous VEGF inhibits vascularization and calcification of the callous [56]. Slit homologue 3 protein (SLIT3) is an axon guidance molecule, which has been shown to induce ECs migration via roundabout homologue (ROBO) signalling [57]. Slit3 mutant mice have reduced Type H vessels and impaired fracture repair, whereas Slit3 overexpression creates a mature callus and increased haematopoiesis during fracture repair [58], suggesting a possible role for type H endothelium in fracture repair (figure 2). The final stage involves the reduction of the fracture callus and normalization of the vasculature. Fracture repair requires increased blood to the site of fracture. In line with this, the aged mice with reduced blood flow to bone exhibit impaired ability to regenerate fractures [59].

The vasculature confers a protective niche for HSCs following chemotherapy, promoting bone and haematopoietic regeneration. Long-term HSCs are associated with arteries and with type H blood vessels which are also referred to as

royalsocietypublishing.org/journal/rsob Open Biol. 9: 190144

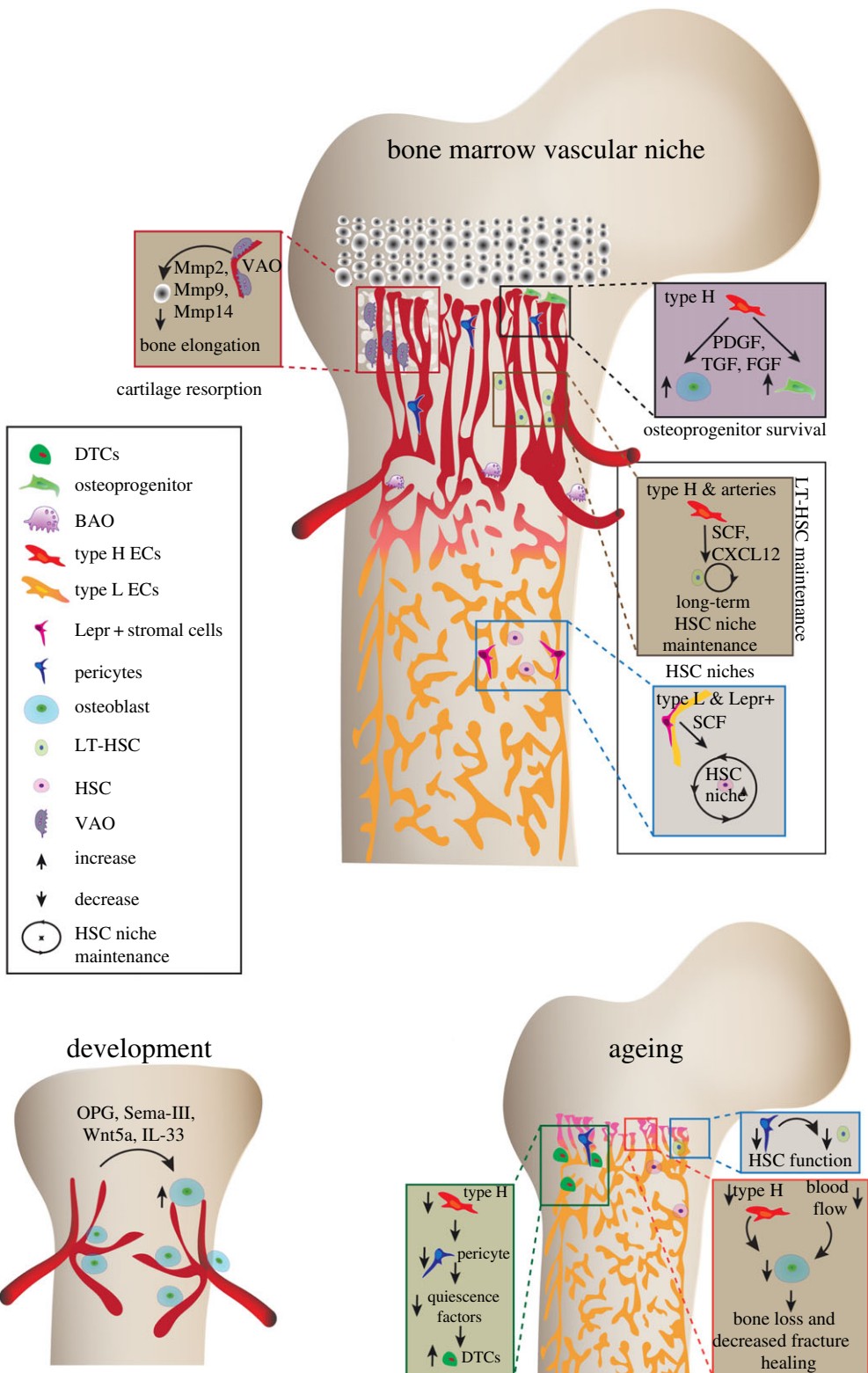

**Figure 1.** Angiocrine crosstalk during bone development, haemostasis and ageing. Displayed are multiple angiocrine factors and their cellular sources that mediate communication between blood vessels, bone cells and haematopoietic cells. Bone development requires blood vessel invasion and osteoprogenitors follow blood vessels. Later, type H blood vessels secrete osteogenic factors and drive the bone formation and bone growth. Further, the proteolytic activity of type H endothelium is required cartilage resorption and directional bone growth. Angiocrine factors derived from different cellular sources maintain HSCs and decline of these cellular sources, particularly, type H and pericytes upon ageing contributes to the declined HSC function. Ageing also leads to enhanced proliferation of DTCs and lowered fracture healing. VAO, vessel-associated osteoclasts; BAO, bone-associated osteoclasts; SCF, stem cell factor; HSC, haematopoietic stem cells; LT, long-term; Mmps, matrix metalloproteinases; PDGF, platelet-derived growth factor; FGF, fibroblast-derived growth factor; TGF, transforming growth factor; CXCL12, C-X-C motif chemokine 12; Lepr+, Leptin receptor; DTC, disseminated tumour cells.

endosteal vessels in some reports [60–62]. The vascular niche is essential to regenerate the HSC population after irradiation [63]. Transplantation of bone marrow ECs following irradiation enhances haematopoiesis and protects radiosensitive tissue [64,65]. Irradiated mice transplanted with bone marrow EC culture conditioned media showed increased survival [64], indicating that angiocrine factors can enhance survival but not compensate for a complete loss of HSCs. Endothelial-specific

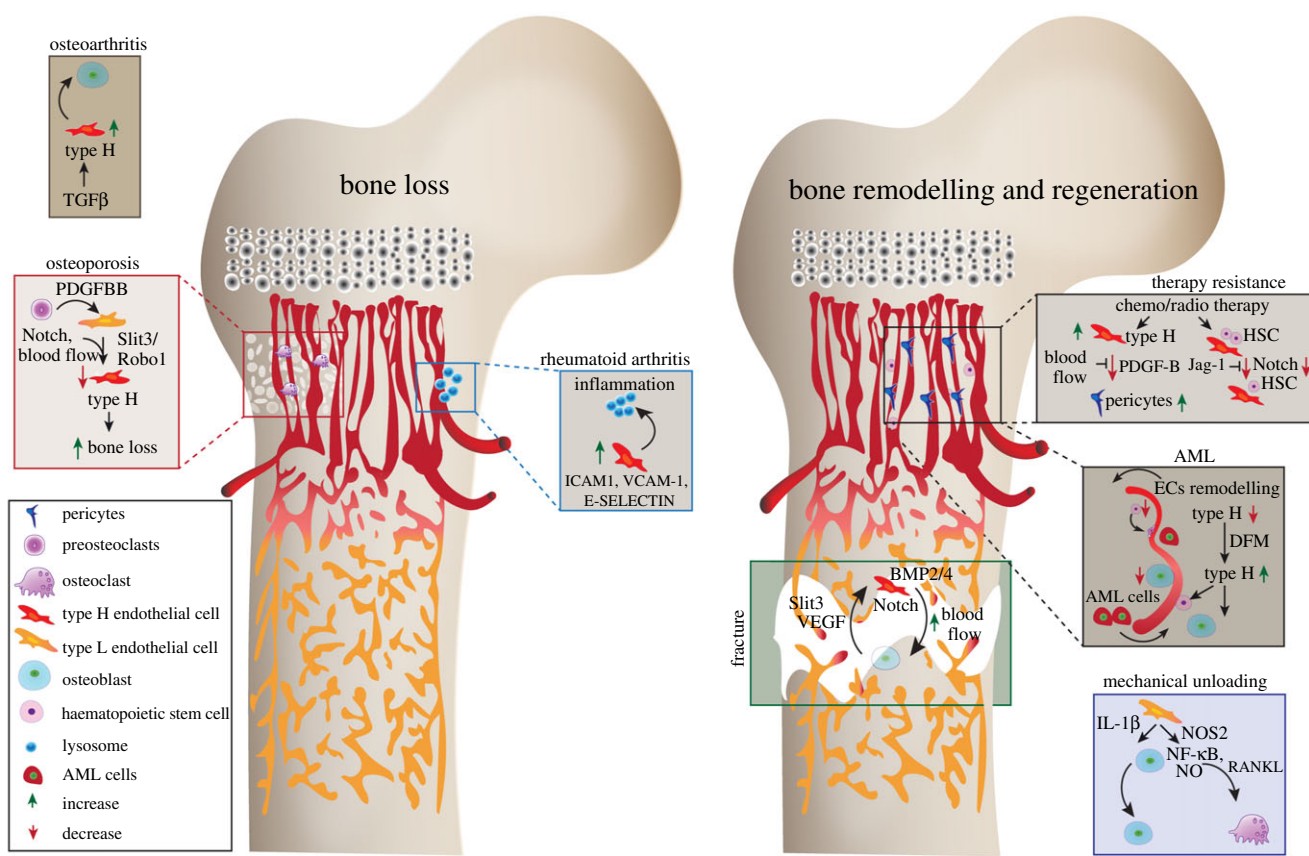

**Figure 2.** Involvement of angiocrine signalling during bone loss, repair and regeneration. Figure illustrating the role of various angiocrine factors, their cellular sources and their influence during radiation and chemotherapy, mechanical loading and also on various pathological conditions like rheumatoid arthritis, OA, inflammation and osteoporosis. The importance of different blood vessel types and associated cells are depicted in the context of bone diseases, repair and regeneration. The angiocrine signalling from the type H ECs plays a crucial role during bone diseases and regeneration. HSC, haematopoietic stem cells; Mmps, matrix metalloproteinases; PDGF, platelet-derived growth factor; FGF, fibroblast-derived growth factor; TGF, transforming growth factor; VEGF, vascular endothelial growth factor; ICAM, intercellular adhesion molecule; VCAM, vascular cell adhesion protein; BMP, bone morphogenetic protein; AML, acute myeloid leukaemia; EC, endothelial cells; NOS, nitric oxide synthase; NO, nitric oxide; IL, interleukin; NF-κB, nuclear factor kappa-light-chain enhancer of activated B cells; RANKL, receptor activator of nuclear factor kappa-B ligand; DFM, deferoxamine mesylate.

deletion of the Notch ligand JAG-1 leads to an impairment in HSC regeneration and increase lethality following irradiation [24]. In addition to Notch signalling, ECs upregulate Fgf-2, Bmp4, Igfbp2 and Angiopoetin-1 to expand the haemopoietic stem progenitor cells (HSPCs) [22,23], indicating these factors may be useful to protect HSC following irradiation. Aged bone marrow ECs impair HSCs and promote a myeloid bias, as demonstrated by transplantation of ECs from the bone marrow of the aged mice into the young recipients [66]. The aged bone marrow has a reduction in PDGFR-β expressing pericytes, which correlates with an expansion of disseminated tumour cells (DTCs). Furthermore, the aged bone marrow secretome promotes proliferation of breast cancer cells in bone. Type H ECs expand in response to radiation and chemotherapy and mediate the regenerative angiogenesis in the bone via blood flow-mediated secretion of PDGF-B, which promotes pericyte expansion [67].

## 4. Dysregulation of angiocrine signalling in bone loss conditions

Osteoporosis is associated with failure to maintain a balance between osteoclasts and osteoblasts, resulting in a loss of bone mass and density. Osteoporosis is predominant in

post-menopausal women and linked to a reduction in the Parathyroid hormone. Osteoporosis mouse models demonstrate a decrease in type H blood vessels [68]. Cathepsin K is a protease expressed by osteoclasts and mediates bone resorption. The Cathepsin K inhibitor prevents degradation of the bone matrix by enhancing PDGF-BB in pre-osteoclasts, which in turn, increases type H blood vessels that promote bone formation through the expansion of osterix-associated cells [68]. Schnurri3 (SHN3) acts cell autonomously to regulate bone formation via osteoblasts while also acting non-cell autonomously by enhancing Slit3/Robo1 to increase the type H blood vessels. The increase in type H blood vessels precedes the increase in bone mass seen in $Shn3^{-/-}$ mice [58], demonstrating the angiocrine crosstalk between type H vessels and osteoblasts. Importantly, type H blood vessels also serve as a biomarker for osteoporosis and bone loss in humans [69]. The physical proximity of type H ECs and osteoblasts supports that notion that these blood vessels secrete a localized gradient of factors that works synergistically with osteoblasts to enhance bone formation [26]. Oestrogen-dependent osteoporosis treatment prevents bone reabsorption, whereas increasing type H blood vessels increases osteoprogenitors [68]. Therefore, the absence of type H vessels may serve as a useful biomarker for disease progression. In addition, the expansion of type H blood vessels during osteoporosis may provide a strategy to increase bone

royalsocietypublishing.org/journal/rsob   *Open Biol.* **9**: 190144

formation, thereby improving the bone quality in this condition. However, the impact on oestrogen on type H blood vessels and the crosstalk of type H blood vessels with tissue during osteoporosis treatment is undetermined. Clinical studies indicate a link between reduced blood flow and bone mineral density in osteoporosis [70]. Further studies in mice demonstrate that reduced blood flow results in a significant reduction of osteoprogenitors [71]. These data suggest a potential therapeutic avenue via increased blood flow and angiogenesis in osteoporosis treatment.

Reduction in mechanical loading leads to a decrease in bone mass [72]. Bone mineral density, volume and blood vessel numbers are unchanged in exercised mice treated with an angiogenesis inhibitor [73]. Capillary density increases in swim exercised mice, suggesting that mechanical loading from muscle is sufficient to promote blood vessel increase [74]. Decreased mechanical loading induces IL-1β in ECs and nitric oxide synthase 2 (NOS2) expression, activating the nitric oxide (NO) and nuclear factor kappa-light-chain enhancer of activated B cells (NF-κB) signalling pathways in osteoblasts, which inhibits osteoblast proliferation. Further, Lipocalin 2 inhibits osteoblast differentiation and activates receptor activator of nuclear factor kappa-B ligand (RANKL) to induce osteoclasts, which combined results in an imbalance of bone turnover resulting in bone loss [25] (figure 2).

## 5. Angiocrine signals during inflammation associated bone loss

Under inflammatory milieu, ECs express BMP-2 [75–77] indicative of their role in bone remodelling. Likewise, ECs produce a glycoprotein–cytokine osteoprotegerin (OPG) in response to a higher concentration of glucose, which inhibits osteoclastogenesis [7]. Production of OPG by ECS under high glucose concentration may minimize bone resorption under diabetic conditions. Interleukin 33 (IL-33), a pro-inflammatory cytokine secreted by Endoglin expressing ECs, is believed to play an essential role in osteogenesis. IL-33 induces the differentiation of bone marrow-derived stromal cells to osteoblasts and increase calcium deposition [12] (figure 2).

Rheumatoid arthritis (RA) is a chronic inflammatory disease leading to bone degradation and joint deformities [78]. Rheumatoid arthritis joints display increased angiogenesis and ECs play a central role in the trafficking of leucocytes into the joint [79]. Additionally, ECs expresses several cytokines and proteases, which enhance inflammation. Intercellular adhesion molecule-1 (ICAM-1), vascular cell adhesion protein 1 (VCAM-1) and E-selectin expressed on ECs stimulate leucocyte and fibroblast migration onto the joint [28–30] (figure 2). Osteoarthritis (OA) displays a similar pathology to RA, with the underlying cause due to mechanical wear and tear. Anterior cruciate ligament transection causes OA like phenotypes, with abnormal bone formation and an increase in angiogenesis in the subchondral bone [80,81]. Increased TGFβ initiates a signalling cascade that recruits mesenchymal stem cells (MSCs) and type H vessels, while exogenous blocking of TGFβ results in a reduction in MSC recruitment and type H vessels, attenuating the OA phenotype [80,81] (figure 2).

## 6. Angiocrine signals in complex and ageing HSC niches

Blood vessels in the skeletal system play crucial roles in blood cell formation by providing nurturing nutrient niche microenvironments for HSCs. Although a common precursor has been suggested for vasculogenesis and primitive haematopoiesis [82], interest to understand the vascular microenvironment in definitive haematopoiesis started with the identification of HSC near blood vessels [61]. Analysing the distribution of HSCs in the whole bone marrow suggest their preferential localization near to the vasculature [83,84]. The recent studies using novel markers such as α-catulin [84] and Hoxb5 [85] also support the existence of blood vessel microenvironment for HSCs. Several blood vessel subtypes and perivascular cell subsets have been reported to interact and regulate HSCs. Interactions of HSCs with different vascular and perivascular cell types in the bone marrow microenvironment is reviewed elsewhere [86–88]. Thus, HSCs reside in specialized, complex niches, which are maintained by a heterogeneous group of cells [20,21,89]. Angiocrine factors secreted by blood vessels regulate HSC self-renewal and quiescence [17,90,91]. Recent improvements with bone imaging methods provide insights into the localization of HSCs within the bone marrow where they frequently localize close to blood vessels [16,84,92]. The stem cell factor (SCF) secreted by type H ECs, sinusoidal ECs and arterial ECs is one of the critical angiocrine factors in HSC maintenance [16]. SCF also plays a role during erythropoiesis and lympho-poiesis [16]. Interleukins (ILs) are a class of cytokines that regulates HSCs and are produced by a wide variety of cells including ECs. In mice, IL-33 alters the HSC fate [12]. IL-33 is known to be secreted during tissue damage; however, its role in HSC niche modification and tissue regeneration is not well studied. Interleukin 7 (IL-7) produced from the perivascular stromal cells maintains a pro-B cell niche associated with HSC niche in the bone marrow [18,19]. IL-7 is necessary for controlling the commitment of lymphoid progenitors to B cells [19]. Perivascular stromal cells, the bone marrow ECs, and osteoblasts produce C-X-C motif chemokine 12 (CXCL12), which is a potent chemokine required for the long-term maintenance and quiescence of the HSC niche [20,89]. The involvement of SCF and CXCL12 in HSC maintenance is depicted in figure 1. Angiocrine factors in HSC regulation and crosstalk between HSCs and endothelium is currently an intense area of study and extensively reviewed [86].

Recent studies highlight the importance of type H capillaries and arteries in maintaining HSCs. The cells forming these vascular structures are strongly positive for SCF. Endothelial Notch activation, which promotes arteriole formation and expansion of type H ECs [27], leads to an increase in platelet-derived growth factor receptor-β (PDGFR-β)/Nestin/ Neuron-glial antigen 2 (NG2)-positive perivascular cells, HSCs and augmented SCF levels, suggesting an enhancement of vascular niche function. Remarkably, EC-specific activation of HIF pathway, which enhanced the number of type H capillaries but had no effect on artery formation and perivascular cells, fails to enhance the number of HSCs. Further detailed analyses of endothelial Notch and HIF signalling in bone indicate that both pathways mediate type H EC expansion independently, whereas only Notch signalling enhances the frequency of HSCs by improving the vascular niche function

[48]. The number of arterioles, type H capillaries, PDGFR-β/NG2-positive perivascular cells and hence SCF levels decline in the ageing bone. This reduced number of arterioles upon ageing is in line with the reported decrease in blood flow to the bone in ageing. The decline in arterioles upon ageing not only provides compelling evidence for the decreased blood circulation in bone but is also likely to induce metabolic changes in aged bones. The decrease in blood flow to bone reduces angiogenesis and type H vessels that lead to a reduction in osteoprogenitor cells and new bone formation [71] (figure 1). The formation of new blood vessels leads to increased blood flow, and tissue perfusion and thereby may lead to alteration in vascular niches, metabolic microenvironments and their functions. Supporting this notion endothelial activation of Notch signalling in aged mice not only lead to increased blood flow to the bone but also improved the vascular niche function and improved the abundance of HSCs [48]. However, long-term repopulation analysis of HSCs from niche-activated aged mice shows that HSC functionality is not improved, which is a consequence of cell-autonomous aspects of HSC ageing such as the accumulation of DNA damage. The EC-derived Notch ligands are able to enhance proliferation and prevent the depletion of long-term HSCs [93]. The activation status of ECs can have a profound influence on modulating the number of long-term HSCs [23]. Taken together, skeletal and HSC ageing is an outcome of complex multicellular vascular microenvironments in combination with HSC intrinsic factors contributing to the age-dependent alterations and loss of stem cells functionality (figure 1).

## 7. Angiocrine crosstalk with tissue during malignancies in bone

Vascular niches in bone hold potential to provide a protective microenvironment for cancer cells via secretion of angiocrine factors [94] Technical advances in high-resolution microscopy, coupled with optimization in processing bone tissue, have allowed the investigation of spatio-temporal dynamics in leukaemia and cancer metastasis mouse models [95]. Acute myeloid leukaemia (AML) presents with disorganized bone marrow vasculature, significant remodelling and reduction of the type H endothelium and trans-endothelial migration of HSC [96,97] (figure 2). In addition, ECs support the growth of AML cells *in vitro* and AML cells localized near ECs show resistance to chemotherapy [98,99]. Inhibition of EC remodelling in AML shows an increase in HSC survival [96]. Lymphoma cells secrete fibroblast growth factor 4 and activate FGFR1 on ECs, upregulating the Notch ligand Jag1 on tumour ECs [100]. This crosstalk establishes the vascular niche as a supportive microenvironment, which in turn supports the proliferation of lymphoma cells in a Notch-dependent manner. ECs in multiple myeloma show an upregulation of genes encoding factors involved in extracellular matrix suggesting that pathological remodelling of the bone marrow microenvironment is dependent on extrinsic factors rather than cell-intrinsic mechanisms

to promote angiogenesis and tumour progression [101]. Increased E-selectin expression on bone ECs enhances bone metastasis through an unknown angiocrine signalling pathway that interacts with the Golgi glycoprotein 1 (Glg1) ligand [32]. DTCs remain in a quiescent state in the bone marrow microenvironment and changes to this microenvironment, such as ageing, leads to reactivation and metastasis [102]. Therefore, bone is one of the most common sites for secondary tumour metastasis. Dormant DTCs are closely associated with the endothelium of the bone marrow [33]. Thrombospondin-1 secreted from ECs creates a stabilized vascular niche, in which DTCs become quiescent [33]. The angiocrine signalling via von Willebrand factor and VCAM1 induce an integrin-mediated chemotherapeutic effect on DTCs, with disruptions in this pathway providing potential treatment modalities to eradicate DTCs [31]. Furthermore, a recent study provides the first insights on the impact of the age-related angiocrine signals in regulating the proliferation and quiescence of tumour cells in bones. Specifically, bone EC-derived PDGF-B signalling regulates dormancy and therapy resistance in bone [67]. However, the cell types and niches promoting the proliferation versus the microenvironments supporting the dormancy of DTCs in bones remains elusive. Likewise, the dissection of the mechanisms leading to the reactivation of the dormant tumour cells in the bone marrow needs further investigation.

## 8. Concluding remarks

It is now becoming increasingly clear that the skeletal vasculature is heterogeneous, and specialized to secrete osteogenesis and haematopoiesis supporting angiocrine factors. Loss of these nurturing angiocrine signals leads to the decline in haematopoietic and mesenchymal stem and progenitor cell function during ageing. Dysregulation of the angiocrine crosstalk drives bone loss diseases and other pathobiological conditions in the skeletal system. Thus, in-depth mechanistic insights into the angiocrine crosstalk within and across heterogeneous bone marrow vascular niches would be of high relevance for designing strategies to manage ageing and pathobiological processes in the skeletal system. Furthermore, the identification of new angiocrine factors and dissecting their role in the bone marrow microenvironment holds the potential to provide new therapeutic targets. Thus, there is an exciting opportunity to unravel a plethora of new players and interactions in complex bone marrow niches, with important implications for basic research and medicine.

Data accessibility. This article has no additional data.

Competing interests. We declare we have no competing interests.

Funding. A.P.K is supported by European Research Council (grant no. StG: metaNiche, 805201), Leuka (grant no. 2017/JGF/001), Medical Research Council (grant no. CDA: MR/P02209X/1), The Royal Society (grant no. RG170326), CRUK Development Fund (grant no. CRUKDF 0317-AK), Kennedy Trust for Rheumatology Research (grant no. KENN 15 16 09) and John Fell Fund, University of Oxford (grant no. 161/061).

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
