## [Reviewer comments · Open Biology]

Review History

RSOB-19-0144.R0 (Original submission)

Review form: Reviewer 1

Recommendation

Accept with minor revision (please list in comments)

Do you have any ethical concerns with this paper?

No

Comments to the Author

The authors present a brief, but compelling overview of the role of angiocrine signals in the skeletal system. The review provides insights on the impact of the angiocrine crosstalk on both bone physiology and pathology, describing how vasculature is involved in maintenance of bone homeostasis, but also in bone loss due to ageing or inflammation and in regulating malignant progression in the bone marrow.

The Concise Review is well written and provides a clear introduction to the topic, referring the

reader to a choice of the key literature for further details. I would only suggest a few minor modifications for clarity and precision:

1) The figures are clear and summarize how angiocrine signals regulate the bone organ. However, they are very compact and writings are difficult to read. I would suggest to enlarge the font and to rearrange the panels vertically, so as to have more space to make each panel bigger and more immediately comprehensible for the reader.

2) On page 9, the sentence starting with "Also, the expansion of Type H blood vessels..." is quite long and would probably benefit from being split in two.

Decision letter (RSOB-19-0144.R0)

27-Aug-2019

Dear Dr Kusumbe

We are pleased to inform you that your manuscript RSOB-19-0144 entitled "Angiocrine signals in bone development, homeostasis and disease" has been accepted by the Editor for publication in Open Biology. The reviewer has recommended publication, but also suggest some minor revisions to your manuscript. Therefore, we invite you to respond to their comments and revise your manuscript.

Please submit the revised version of your manuscript within 7 days. If you do not think you will be able to meet this date please let us know immediately and we can extend this deadline for you.

1) A text file of the manuscript (doc, txt, rtf or tex), including the references, tables (including captions) and figure captions. Please remove any tracked changes from the text before submission. PDF files are not an accepted format for the "Main Document".

2) A separate electronic file of each figure (tiff, EPS or print-quality PDF preferred). The format

should be produced directly from original creation package, or original software format. Please note that PowerPoint files are not accepted.

3) Electronic supplementary material: this should be contained in a separate file from the main text and meet our ESM criteria (see <http://royalsocietypublishing.org/instructions-authors#question5>). All supplementary materials accompanying an accepted article will be treated as in their final form. They will be published alongside the paper on the journal website and posted on the online figshare repository. Files on figshare will be made available approximately one week before the accompanying article so that the supplementary material can be attributed a unique DOI.

Online supplementary material will also carry the title and description provided during submission, so please ensure these are accurate and informative. Note that the Royal Society will not edit or typeset supplementary material and it will be hosted as provided. Please ensure that the supplementary material includes the paper details (authors, title, journal name, article DOI). Your article DOI will be 10.1098/rsob.2016[last 4 digits of e.g. 10.1098/rsob.20160049].

4) A media summary: a short non-technical summary (up to 100 words) of the key findings/importance of your manuscript. Please try to write in simple English, avoid jargon, explain the importance of the topic, outline the main implications and describe why this topic is newsworthy.

Images

Data-Sharing

It is a condition of publication that data supporting your paper are made available. Data should be made available either in the electronic supplementary material or through an appropriate repository. Details of how to access data should be included in your paper. Please see <http://royalsocietypublishing.org/site/authors/policy.xhtml#question6> for more details.

Data accessibility section

Sincerely,

The Open Biology Team
<mailto:openbiology@royalsociety.org>

Reviewer(s)' Comments to Author:

Referee:

Comments to the Author(s)

The authors present a brief, but compelling overview of the role of angiocrine signals in the skeletal system. The review provides insights on the impact of the angiocrine crosstalk on both bone physiology and pathology, describing how vasculature is involved in maintenance of bone homeostasis, but also in bone loss due to ageing or inflammation and in regulating malignant progression in the bone marrow.

The Concise Review is well written and provides a clear introduction to the topic, referring the reader to a choice of the key literature for further details. I would only suggest a few minor modifications for clarity and precision:

- 1) The figures are clear and summarize how angiocrine signals regulate the bone organ. However, they are very compact and writings are difficult to read. I would suggest to enlarge the font and to rearrange the panels vertically, so as to have more space to make each panel bigger and more immediately comprehensible for the reader.
- 2) On page 9, the sentence starting with "Also, the expansion of Type H blood vessels..." is quite long and would probably benefit from being split in two.

Author's Response to Decision Letter for (RSOB-19-0144.R0)

See Appendix A.

Decision letter (RSOB-19-0144.R1)

04-Sep-2019

Dear Dr Kusumbe

We are pleased to inform you that your manuscript entitled "Angiocrine signals in bone development, homeostasis and disease" has been accepted by the Editor for publication in Open Biology.

Sincerely,

The Open Biology Team
mailto:openbiology@royalsociety.org

Appendix A

Reviewer(s)' Comments to Author:

We are grateful to the editor and reviewer for their time. We have addressed the comments as suggested. A detailed point-by-point reply is provided below.

Referee:

Comments to the Author(s)

The authors present a brief, but compelling overview of the role of angiocrine signals in the skeletal system. The review provides insights on the impact of the angiocrine crosstalk on both bone physiology and pathology, describing how vasculature is involved in maintenance of bone homeostasis, but also in bone loss due to ageing or inflammation and in regulating malignant progression in the bone marrow.

The Concise Review is well written and provides a clear introduction to the topic, referring the reader to a choice of the key literature for further details. I would only suggest a few minor modifications for clarity and precision:

We are grateful to the reviewer for the positive comments and valuable suggestions.

1) The figures are clear and summarize how angiocrine signals regulate the bone organ. However, they are very compact and writings are difficult to read. I would suggest to enlarge the font and to rearrange the panels vertically, so as to have more space to make each panel bigger and more immediately comprehensible for the reader.

We thank the reviewer for this important comment. We have increased the font size and rearranged the figures so that these are bigger.

2) On page 9, the sentence starting with “Also, the expansion of Type H blood vessels...” is quite long and would probably benefit from being split in two.

We thank the reviewer for bringing this to our attention. As suggested we have now split this sentence in two.